# Spatiotemporal Analysis of the Population Risk of Congenital Microcephaly in Pernambuco State, Brazil

**DOI:** 10.3390/ijerph17030700

**Published:** 2020-01-21

**Authors:** Neal D. E. Alexander, Wayner V. Souza, Laura C. Rodrigues, Cynthia Braga, André Sá, Luciana Caroline Albuquerque Bezerra, Celina Maria Turchi Martelli

**Affiliations:** 1Department of Infectious Disease Epidemiology, London School of Hygiene and Tropical Medicine, London WC1E 7HT, UK; laura.rodrigues@lshtm.ac.uk; 2Centro de Pesquisas Aggeu Magalhães, Fundação Oswaldo Cruz, Recife PE 50740-465, Brazil; wayner@cpqam.fiocruz.br (W.V.S.); braga@cpqam.fiocruz.br (C.B.); andre.sa@cpqam.fiocruz.br (A.S.); turchicm@gmail.com (C.M.T.M.); 3State Health Department of Pernambuco, Recife PE 50751-530, Brazil; lua_cad@yahoo.com.br

**Keywords:** congenital microcephaly, Zika, Brazil, arbovirus

## Abstract

Since an outbreak in Brazil, which started in 2015, Zika has been recognized as an important cause of microcephaly. The highest burden of this outbreak was in northeast Brazil, including the state of Pernambuco. The prevalence of congenital microcephaly in Pernambuco state was estimated from the RESP (Registro de Eventos em Saúde Pública) surveillance system, from August 2015 to August 2016 inclusive. The denominators were estimated at the municipality level from official demographic data. Microcephaly was defined as a neonatal head circumference below the 3rd percentile of the Intergrowth standards. Smoothed maps of the prevalence of microcephaly were obtained from a Bayesian model which was conditional autoregressive (CAR) in space, and first order autoregressive in time. A total of 742 cases were identified. Additionally, high and early occurrences were identified in the Recife Metropolitan Region, on the coast, and in a north–south band about 300 km inland. Over a substantial part of the state, the overall prevalence, aggregating over the study period, was above 0.5%. The reasons for the high occurrence in the inland area remain unclear.

## 1. Introduction

In November 2015, an epidemic of congenital microcephaly, associated with the Zika virus, was declared a Public Health Emergency of National Importance by the Brazilian Ministry of Health [1]. In the following year, the World Health Organization declared a Public Health Emergency of International Concern [2]. On the basis of the Bradford Hill criteria, a causal link has been inferred [3], and the Centers for Disease Control and Prevention of the USA has concluded that ‘There is now scientific consensus that Zika virus infection during pregnancy is a cause of microcephaly’ [4]. The main transmission route of Zika is by mosquitoes of the genus *Aedes* [5].

Causes of congenital microcephaly include chromosomal abnormalities (e.g., trisomy 21), monogenic disorders (e.g., Cornelia de Lange syndrome), in utero exposure to teratogens (e.g., alcohol, heavy metals), as well as maternal infections (e.g., toxoplasmosis, rubella, cytomegalovirus) [6]. The atypical pattern of microcephaly cases initially described in the Northeast of Brazil drew the world’s attention to a distinct pattern of birth defects, which was subsequently designated congenital Zika syndrome (CZS) [7]. The clinical features include microcephaly, abnormal brain development, limb contractures, eye abnormalities, brain calcifications, and other neurologic findings. In Brazil, 4200 confirmed or possible congenital cases related to Zika infection were reported from 2015 to 2017; 60% of them in the northeast region. Although representing a sharp reduction in incidence, the 73 cases related to Zika infection reported in 2018–2019 indicate ongoing transmission [8]. There are still many knowledge gaps related to the progression of the disease and the risk of the transmission of Zika infection in tropical settings.

Previous spatial and temporal analyses of microcephaly and congenital Zika syndrome, have considered all of Brazil, Pernambuco, or Recife [9,10,11,12,13,14]. The contribution of the current analysis is to describe the patterns of microcephaly in Pernambuco in 2015 and 2016 at a fine scale in terms of both space (municipality) and time (month).

## 2. Materials and Methods

### 2.1. Setting

Pernambuco is a northeastern state of Brazil (Figure 1). It extends approximately 700 km inland from the Atlantic Ocean and has 8.8 million inhabitants, 42% of which reside in the Metropolitan Region of Recife (RMR), on the coast, which consists of 14 municipalities including Recife proper (outlined in red in Figure 1) [15].

### 2.2. Microcephaly Surveillance

In October 2015, the State Secretariat of Health in Pernambuco launched the first online registry to report microcephaly and STORCH (syphilis, toxoplasmosis, other, including HIV, rubella, cytomegalovirus, and herpes simplex) due to the excess of these events at a state level. This initiative was followed in November 2015 by a national-level system with an electronic database for the notification of cases and deaths with suspected relation to Zika infection: RESP (Registro de Eventos em Saúde Pública or Public Health Events Registry) [16]. Health facilities were advised to notify any suspected neonatal microcephaly cases to the RESP system, including those born before its inauguration, if possibly related to maternal infection during pregnancy [17]. We extracted data from August 2015 to August 2016 inclusive, which covers the main peak of the outbreak [10], and the sample size was not pre-determined. The head circumference was included in the source data, and microcephaly was defined, for the purpose of the current study, as the neonatal head circumference below the 3rd percentile of the Intergrowth standards, which are specific for sex and gestational age [18].

### 2.3. Live Birth Denominator

The number of live births in each month in each municipality was estimated from the number of live births in the corresponding month and municipality in 2013, from the national Live Birth Information System (SINASC, Sistema de Informações sobre Nascidos Vivos) of the Ministry of Health. Some municipalities had zero births for some months, so the denominator was smoothed, by taking it to be the average of the original value with the two values from months adjacent in time and then rounding to the nearest integer. 

### 2.4. Statistical Analysis and Reporting

We used a spatiotemporal Bayesian smoothing of the proportion of live births with microcephaly, using the WinBUGS software [19]. The model follows those used by Lawson (Chapter 12) [20] and Knorr-Held [21]. More specifically, the proportion with microcephaly is assumed to follow a binomial distribution with the log-odds, including structured spatial and temporal components, an unstructured spatial component, and a space-time interaction. The structured components are conditional autoregressive (CAR) in space and first order (1 month) autoregressive in time. The CAR neighbourhood matrix is determined by whether or not each pair of municipalities have any common borders.

The analysis includes the 184 contiguous municipalities of Pernambuco (S1 code), excluding Fernando do Noronha, which is archipelago of 21 small islands, 350 km offshore. We also estimated, for each municipality, the month in which the prevalence first passed 1 in 500 births, and the prevalence over the complete study period. Parameters were estimated from 10,000 iterations after 5000 burn-in iterations.

A map of the municipalities of Pernambuco was obtained in a shapefile format from the website of the IBGE (Brazilian Institute of Geography and Statistics, https://www.ibge.gov.br/geociencias/downloads-geociencias.html). Choropleth maps of the raw data, and the results of the CAR model, were plotted using this shapefile in the R software [22]. Maps of the countries of South America, and of the states of Brazil, were made using the R packages dplyr, ggmap, maps maptools, raster, and rgdal. The reporting is in accordance with the RECORD guidelines [23] (S2 checklist).

### 2.5. Ethics

Pernambuco Health State Department granted permission to use the anonymous data for research purposes. In addition, this analysis was approved by the ethics committee of the London School of Hygiene and Tropical Medicine (reference 16111). No names were present in the dataset, and addresses and telephone numbers were removed before analysis.

## 3. Results

The analysis included 742 cases, which were selected as shown in Figure 2. Maps of raw prevalence are shown in Figure 3. Of the 184 municipalities, 49 (27%) had no case reported during the period; the median was 2 cases and the upper quartile 4 cases. The 14 municipalities of the Recife Metropolitan Region had 38% of the cases and 40% of the expected births.

In terms of time, the peak was in November 2015. In terms of space, as well as high proportions in the Recife Metropolitan Region, high prevalences occur in a north–south band about 300 km inland. Figure 4 and Figure 5 show results from the spatiotemporal CAR model used to smooth the data. Figure 4 shows the month in which the fitted prevalence first passed 1 in 500 births. This occurred early in some coastal municipalities in the RMR but also, notably, in the same north–south inland band. Figure 5 shows corresponding patterns of prevalence over all the study period, which generally reflect those of Figure 4. Over a substantial part of the state, this overall prevalence was over 0.5%.

## 4. Discussion

These results show that the outbreak of microcephaly in Pernambuco state, as reported in the RESP surveillance system, was not only concentrated in the coastal Recife Metropolitan Area. In a band running north–south across the state, approximately 300 km from the coast, the occurrence of microcephaly also increased early and had a high overall prevalence within the outbreak. Such an inland high risk area was not visible in the kernel density estimates of microcephaly in a previous national-level analysis [10], although it was in corresponding estimates of possible Zika virus infections in pregnant women [10] and in a recent national-level analysis of microcephaly [14]. The current analysis shows that prevalence peaked early as well as high in this inland region.

A similar area was found to be environmentally suitable for the transmission of Zika by *Aedes* [24]. However we cannot rule out that this inland peak resulted, rather, from maternal Zika infection acquired elsewhere, especially bearing in mind the lag between infection and birth. Indeed, one limitation of the current analysis is a lack of information on travel. On the other hand, the environmental explanation could be considered more parsimonious, given that there is a distinct inland peak, not a gradient from the coast, which might be expected if travel were the explanation. Zika can also be transmitted sexually, although this is not considered the principal route in tropical and subtropical areas due to the abundance of *Aedes* mosquitoes.

De Oliveira et al. [10] showed the peak risk of microcephaly in the northeast region to be 0.57% in December 2015. By contrast, in terms of peak prevalence from the current data, the median over municipalities was much higher, at 1.5% (inter-quartile range 1.3–1.9%). In terms of the prevalence over the whole study period, the findings in the current paper (median prevalence across municipalities of 0.41%) are more in accordance with those of another study in Recife city, which found the overall prevalence, from August 2015 to July 2016, to be 0.61% [13]. On the other hand, it is lower than the 0.74% found by de Araújo et al. [25], although this may be explained by their study period being from January to December 2016. Although we identified an inland area with an early peak of microcephaly, in terms of numbers the highest burden was in the RMR, and remains so in terms of current disability. At the national level, Zika infection continues to pose an increased risk for adverse pregnancy outcomes, such as preterm birth, fetal death, and stillbirth, and congenital Zika syndrome (CZS) [8].

A limitation of the current study is the possible error in ascertaining microcephaly, due to the difficulty in taking the measuring head circumference [26] and possible bias in a time of heightened awareness of the condition. The surveillance protocol [17] called for congenital microcephaly to be reported if there was a relation to maternal infection Zika infection, which could be difficult to assess. Surveillance of the head circumference, were it to be done irrespective of other signs, would “inevitably include some children who are constitutionally small but otherwise clinically normal” [27]. However, the prevalence in the current study is comparable to that found by de Oliveira et al. on the national level [10]. In South America, causes of microcephaly prior to the Zika outbreak included chromosomal syndromes (e.g., Patau), monogenic syndromes (e.g., Meckel), associated neural malformations (e.g., encephalocele), and embryopathies, which were largely related to infections (e.g., cytomegalovirus and toxoplasmosis) and less frequently to alcohol [28]. However, as is the case with surveillance in the United States, for example [27], the majority of cases did not have a documented cause and, in the current study, individual etiological investigation is only partially available. Suggestions that maternal vaccination, or the insecticide pyriproxyfen, could be risk factors for microcephaly have not been borne out [25,29,30].

The descriptive methodology of the current paper for the spatial and temporal distribution of the microcephaly cases does not include the evaluation of the association with congenital Zika infection, and doing so would require another set of statistical methods. Such an association has been well established in case-control and cohort studies in several settings [3,25]. Also, the data on Zika in Pernambuco in the study time frame have limitations in that the surveillance systems were still being developed, and diagnostic laboratory tests were limited [14]. However, at the descriptive level, the spatial and temporal patterns of microcephaly shown here coincide with those of Zika in Brazil shown by, for example, Brady et al. from December 2015 onwards [14], and Paixão et al. in an interrupted time series analysis by region [31].

The number of microcephaly cases is often small when broken down by municipality and month, so sporadic, relatively high, values could be misleading when mapping crude risks. Hence we used a smoothing methodology to give more trustworthy estimates [20,32]. On the other hand, a limitation of the maps is that those municipalities with high populations, typically on the coast, tend to have a smaller area, and hence less graphical prominence than those inland. Finally, the statistical analysis is descriptive, and there is no attempt to define clusters in terms of precise geographical extent or statistical significance.

## 5. Conclusions

This study emphasizes the burden of microcephaly during the outbreak in Pernambuco, with estimates higher than in some previous studies, and with high risk in an inland region of the state for reasons which are yet to be clarified. There was a high spatial dispersion across municipalities of congenital microcephaly in a relatively short time. These findings have important public health implications for monitoring the high-risk municipalities (possible hot spots), and investigating the previous mobility (travel or commuting) of Zika-infected pregnant women. This is to understand infectious spread, target vector control measures, and to enhance pre-natal services. We also support the clinical and laboratory follow-up of microcephaly cases, to enhance the ascertainment of CZS by the implemented surveillance system.

## Figures and Tables

**Figure 1 ijerph-17-00700-f001:**
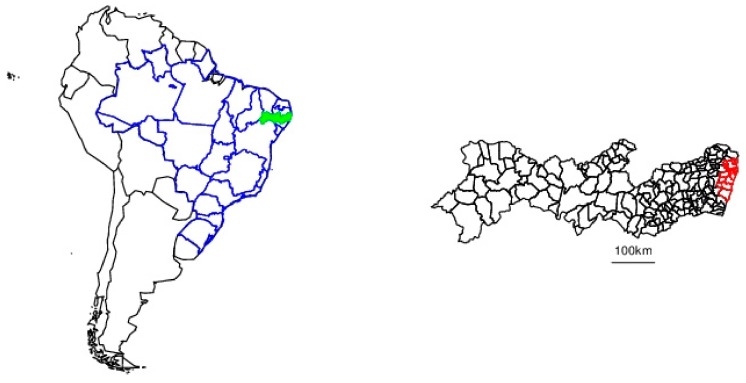
Pernambuco. Left panel: South America, with the national boundaries in black, and the Brazilian state boundaries in blue. Pernambuco is shown in green. Right panel: municipalities of Pernambuco, with those of the Recife Metropolitan Region being shown in red.

**Figure 2 ijerph-17-00700-f002:**
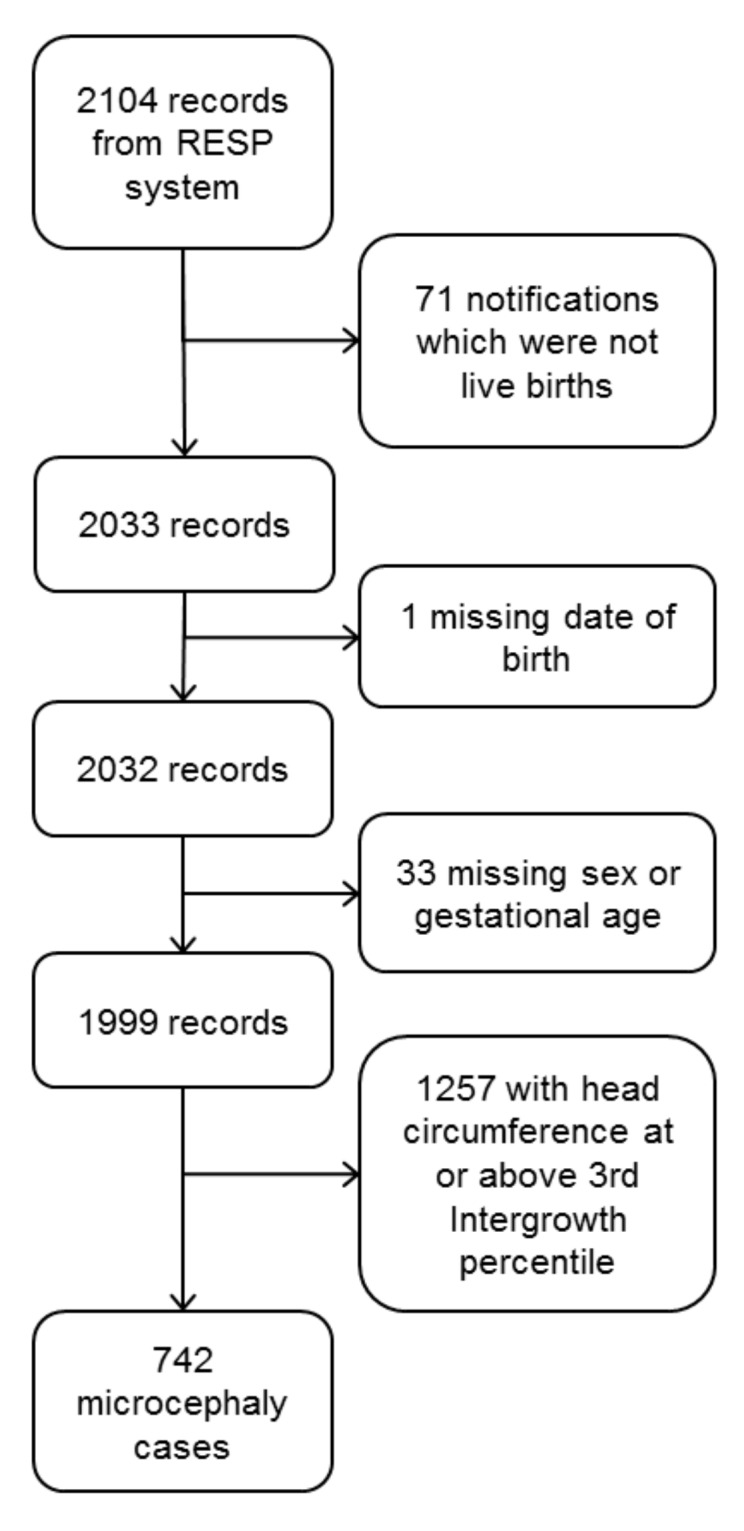
Flowchart of the inclusion of birth records from the Registro de Eventos em Saúde Pública (RESP) system.

**Figure 3 ijerph-17-00700-f003:**
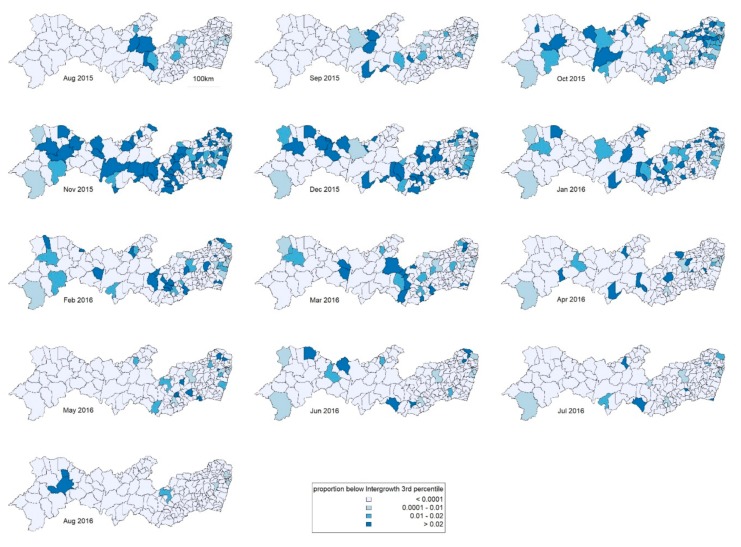
Maps of raw microcephaly prevalence over time.

**Figure 4 ijerph-17-00700-f004:**
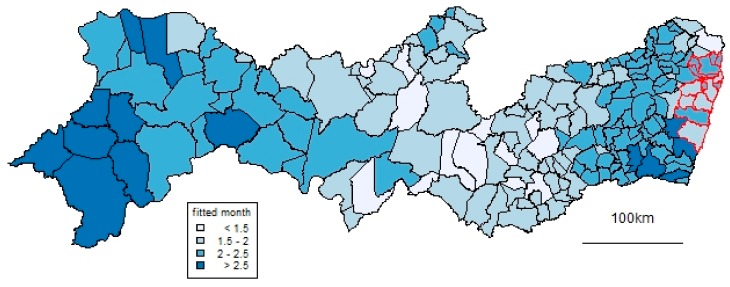
Month when the smoothed prevalence of microcephaly first reached 1 in 500. Lighter colors indicate that this threshold was reached earlier. Month 1 means August 2015, and so on.

**Figure 5 ijerph-17-00700-f005:**
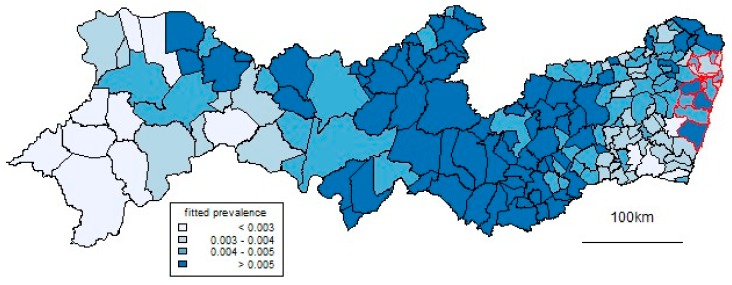
Fitted prevalence of congenital microcephaly over the study period.

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
