# Peer review of "Spatiotemporal Analysis of the Population Risk of Congenital Microcephaly in Pernambuco State, Brazil"

_ijerph, 2020, doi:10.3390/ijerph17030700_

Round 1

Reviewer 1 Report

The article addresses an important topic of public health interest in Brazil seeking to measure the prevalence of microcephaly and its temporal and spatial distribution in the state of Pernambuco. However, it does not show any association between microcephaly and Zika.

Perhaps the title should be reformulated because it is not in line with the objective, the study does not show that microcephaly is due to Zika Virus.

The introduction, although incisive, is very brief, the subject deserves a more robust framework.

The methodology is adequate to the results presented by the authors.

The results and their analysis is only descriptive, although the authors recognize this limitation in the discussion. It seems to me that it might be appropriate to study the association between microcephaly and Zika and to see if the differences found in prevalence are significant in different regions and different periods and try to understand what factors may justify these differences.

The conclusion is very synthetic, the authors should present some implications of their findings in the clinic and public health policy of the state of Pernambuco.

Author Response

1            The article addresses an important topic of public health interest in Brazil seeking to measure the prevalence of microcephaly and its temporal and spatial distribution in the state of Pernambuco. However, it does not show any association between microcephaly and Zika.

   Response:

   Thanks for the positive comment.  We have tried to clarify that our conclusions about the association between microcephaly and Zika are limited.  In particular we have modified the title as suggested (point 2 below) and, in the discussion, clarified that the paper does not include formal analysis of any association with Zika.

2            Perhaps the title should be reformulated because it is not in line with the objective, the study does not show that microcephaly is due to Zika Virus.

   Response:

   Agreed.  We have deleted “Attributed to Zika Virus Infection” from the title. 

3            The introduction, although incisive, is very brief, the subject deserves a more robust framework .

   Response:

   We agree that it was overly brief.  We now contextualize the topic by describing, in the introduction other causes of microcephaly, the course of the recent increase in microcephaly, the evidence for its association with Zika, and its impact in terms of Congenital Zika Syndrome.

4            The methodology is adequate to the results presented by the authors.

   Response:

   Thanks for the positive comment.  (No resulting changes.)

5            The results and their analysis is only descriptive, although the authors recognize this limitation in the discussion. It seems to me that it might be appropriate to study the association between microcephaly and Zika and to see if the differences found in prevalence are significant in different regions and different periods and try to understand what factors may justify these differences.

   Response:

   We agree that this was a gap in the paper.  We now address this in the discussion.  We do not attempt to include more source data on Zika, which would be required for formal analysis of the association, because this is beyond our objectives and would enlarge the paper multifold. 

6            The conclusion is very synthetic, the authors should present some implications of their findings in the clinic and public health policy of the state of Pernambuco.

   Response:

   We agree with this constructive suggestion.  We have expanded the “Conclusions” section to describe the public health implications. 

Reviewer 2 Report

The current manuscript is fascinating, provides data for the occurrence of the microcephaly in the Pernambuco state of Brazil. It is not clear that Zika causes the microcephaly described here. Zika virus is one of the causes of microcephaly, but there are several other causes of the disease. Data were not collected to show these children born had Zika viral infection. Pernambuco state is northeast of the country, which get affected by the Zika infection between December to March due to the increase in the vector in the rainy season. In the cases of microcephaly, the effects of the disease might be delayed due to developmental reasons. However, the present data seems incomplete without data for Zika positivity. Microcephaly in the inland may be due to rubella, toxoplasmosis, or cytomegalovirus infection. It may also be caused by malnutrition, drug abuse, etc.

Author Response

1            The current manuscript is fascinating, provides data for the occurrence of the microcephaly in the Pernambuco state of Brazil. It is not clear that Zika causes the microcephaly described here. Zika virus is one of the causes of microcephaly, but there are several other causes of the disease. Data were not collected to show these children born had Zika viral infection. Pernambuco state is northeast of the country, which get affected by the Zika infection between December to March due to the increase in the vector in the rainy season. In the cases of microcephaly, the effects of the disease might be delayed due to developmental reasons. However, the present data seems incomplete without data for Zika positivity. Microcephaly in the inland may be due to rubella, toxoplasmosis, or cytomegalovirus infection. It may also be caused by malnutrition, drug abuse, etc.

   Response:

Thanks for the positive comments. 

We agree about the association and, as mentioned also for the other reviewer, we have modified the title, and added material to discussion to clarify that the paper does not include formal analysis of any association with Zika.  Also, despite the previous evidence for a causal link between Zika and microcephaly, we agree that it was an important omission not to address other causes, and we now do so in the introduction and discussion. 

Round 2

Reviewer 1 Report

The authors made some changes to the article according to the suggested suggestions.
However, the study of the association of the disease with the risk factors is not done. This aspect would significantly enrich the scientific content of the article.

Author Response

We have expanded the discussion to further address possible risk factors, notably Zika.  However, for reasons now described more fully in the discussion, we have not carried out statistical analysis of the association between microcephaly and Zika. 

Reviewer 2 Report

N/A

Author Response

We’re happy to see that this reviewer found the R1 version acceptable.

Round 3

Reviewer 1 Report

The answers of the authors are satisfactory.